# The Importance of Perceived Relevance: A Qualitative Evaluation of Patient’s Perceptions of Value and Impact Following a Low-Intensity Group-Based Pain Management Program

**DOI:** 10.3390/medicina57010046

**Published:** 2021-01-07

**Authors:** Joshua W. Pate, Elizabeth Tran, Seema Radhakrishnan, Andrew M. Leaver

**Affiliations:** 1Graduate School of Health, University of Technology Sydney, Sydney 2007, Australia; 2Westmead Hospital Pain Management Centre, Sydney 2145, Australia; elizabeth.tran@health.nsw.gov.au (E.T.); seema.radhakrishnan@health.nsw.gov.au (S.R.); andrew.leaver@sydney.edu.au (A.M.L.); 3Faculty of Health Sciences, University of Sydney, Sydney 2141, Australia

**Keywords:** persistent pain, pain management program, qualitative interviews, perceived values, perceived impacts

## Abstract

*Background and objectives:* Limited evidence exists exploring perceptions of which aspects of a pain management program are perceived as valuable and impactful. The aim of this study was to explore patient beliefs about which aspects of a pain management program were valued and/or had perceived impact. *Materials and Methods:* One-on-one structured interviews were conducted with 11 adults three months after their completion of the Spark Pain Program at Westmead Hospital, Sydney, Australia. Concepts in the transcripts were inductively identified and explored, utilizing thematic analysis to better understand their relevance to the study aim. *Results*: Four themes emerged: (1) “The program overall was positive, but…”; (2) “I valued my improved knowledge and understanding of pain, but…”; (3) “I valued the stretching/relaxation/pacing/activity monitoring”; and (4) “I valued being part of a supportive and understanding group”. Participants reported that they liked being treated as an individual within the group. A lack of perceived personal relevance of key messages was identified in some participants; it appears that patients in pain programs must determine that changes in knowledge, beliefs, and attitudes are personally relevant in order for the changes to have a significant impact on them. *Conclusions:* This study provides new insights into aspects of a pain management program that were perceived as valuable and impactful, areas that “missed the mark”, and hypotheses to guide the implementation of service delivery and program redesign.

## 1. Introduction

Persistent pain creates an enormous biopsychosocial burden. The most frequent cause of years lived with disability globally is persistent low back pain, and other frequent causes include chronic neck pain, osteoarthritis, musculoskeletal disorders, migraine, and medication overuse headache [1]. Persistent pain is associated with depressive symptoms and worsened perceptions of general health, activities of daily living, and relationships [2,3]. Approximately 15.5% of work absence is due to back pain [4], and in terms of financial impacts, a 2018 report estimates that overall annual costs of persistent pain in Australia exceed AU$73 billion (AU$139 billion if reductions in quality of life are included) [5].

To address this burden, multidisciplinary pain management programs are the gold standard approach for the treatment of persistent pain, targeting changes in beliefs and behaviors, and encouraging acceptance, coping, and self-management. Current guidelines recommend non-pharmacological and non-invasive management, including patient education, exercise therapy, and advice to remain active [6]. Multidisciplinary biopsychosocial rehabilitation interventions were found to be more effective than usual care in decreasing pain and disability in adults with chronic low back pain in a 2015 meta-analysis [7]. Despite the complexity of implementing behavior-change interventions, a recent review agreed with these findings and additionally found that exercise and multidisciplinary rehabilitation were associated with an increased likelihood of returning to work [8]. Further, Groot et al. (2019) recently confirmed that long-term benefits of reduced healthcare utilization extend beyond 5 years for selected and motivated patients [9]. In Australia, pain programs all use a standard set of outcome measures [10], are becoming increasingly common [11], and are categorized by their duration as low intensity (6–24 h), medium intensity (24–60 h), or high intensity (60–120 h) [12]. In terms of targeting treatments and optimizing cost-benefit analyses to maximize value and impact, lower intensity pain management programs have been developed [12] for people with persistent pain and relatively lower psychological distress and disability [13,14,15].

Limited evidence exists exploring perceptions of which aspects of a pain management program are valuable and impactful, which could guide the implementation of service delivery and program redesign. Qualitative studies, which can achieve this purpose, have previously concentrated on other aspects including, but not limited to: patient values in the years following a high-intensity program [16], the patient’s perceived barriers and facilitators to self-management [17,18,19], the patient’s perceptions of themselves [20] and of changes that have occurred after a program [16,21], broader perceptions of pain treatments and life [22], navigating chronic pain healthcare systems [23,24], successful strategies used outside of a pain management program [25], perspectives on opioids [26,27], and patient’s expectations about a program [28]. Understanding more about participant perceptions could improve the acceptance of active evidence-based therapies like pain management programs [29]. Using a foundation of pain science education [30,31,32,33], the key aims within these programs are to improve understanding, lessen fears and anxiety, challenge unhelpful beliefs and encourage behavioral change. The patient-centered design [34] and re-design of pain management programs could potentially be improved via an enhanced understanding of patients’ perceptions of both the value and impact of different components of a program. Therefore, the aim of this study was to explore patient beliefs about which aspects of a pain management program were valued and/or had perceived impact. 

## 2. Materials and Methods

The study was approved by the Human Research Ethics Committee of Western Sydney Local Health District on 17 April 2019 (Ref: 2019/ETH00462).

### 2.1. Design

This qualitative study utilized one-on-one structured interviews, lasting approximately 10 to 20 min, of adults three months after their completion of a low-intensity multidisciplinary pain management program, as part of the departmental quality improvement program. The ‘Spark Pain Program’ runs for 2 h, once per week, for 6 consecutive weeks at Westmead Hospital Pain Management Centre in Sydney, Australia. This 12-h program containing 6–12 participants is jointly coordinated by a senior physiotherapist and clinical psychologist. Each week involves pain science education based on published target concepts [35] (20 min), a relaxation activity (10 min), a range of whole-body exercises (30 min), problem-solving activities (30 min), gentle and relaxed stretching (20 min) and a goal-setting activity (10 min), with a pain specialist providing three brief 5-min sessions on the role of physicians and medications as part of the education components. The study was designed and reported according to the 32-item Consolidated Criteria for Reporting Qualitative Health Research (COREQ) (Appendix A) [36], except that repeat interviews were not conducted due to limited study resources. Concepts in the transcripts were inductively identified and recorded utilizing thematic analysis [37,38,39]. A coding structure was then iteratively developed and refined until the aims of the study were achieved. Themes and sub-themes were developed from data analysis and synthesis.

### 2.2. Participants

The study population consisted of participants of the Spark Pain Program between October 2018 and June 2019. Patients are eligible for the Spark Pain Program if they are aged >18 years and have had pain for more than 3 months. Personalized emails and phone calls were used to invite consecutive participants to take part in a follow-up interview three months after their completion of the Spark Pain Program. Participants were interviewed until “information power” was established [40,41]. Interviews were conducted between January and September 2019.

### 2.3. Interview Script

An interview script was developed (Table 1) to explore the patient’s perceptions of the value and impact of the Spark Pain Program. Pilot testing on two healthy, pain-free adults, discussions with education experts, and further review by the investigators resulted in the final script. Following an iterative process, the findings from interviews informed the focus of subsequent interviews until data saturation was reached.

### 2.4. Data Collection

Individual structured interviews were conducted by a research assistant (E.T.). The research assistant was a health professional who had no previous relationship with the participants. Interviews took place at a time chosen by the participant with no one else present. Interviews were, on average, 11 min (range = 6–18) and conducted via telephone. Interviews were audio-recorded, transcribed, and de-identified by the research assistant. Participants were invited to review the transcripts for accuracy before analysis. Data saturation was defined as when no additional new information was attained with additional interviews regarding the list of codes contributing to themes and subthemes. Data collection was stopped when all four researchers agreed that saturation had taken place.

### 2.5. Data Analysis

Audio recordings from interviews were transcribed verbatim by the research assistant (E.T.) and checked for accuracy by another researcher (J.W.P.). The data was then thematically analyzed and synthesized [39]. 

Because three of the researchers (J.W.P., S.R., and A.M.L.) are also pain-related clinicians, it was impossible not to have preconceptions about patient perceptions. Therefore, contrary to the original formulations of theories such as grounded theory, we a priori had consulted relevant research. The constant comparison method of grounded theory [42] was used, where each interview transcript was coded to identify salient themes, and the coding structure was modified as new themes emerged. Data were independently coded line-by-line by all four researchers. The team of four researchers analyzing the data met fortnightly throughout data collection, and the thematic analysis determined when data saturation was reached. Analytical themes were inductively developed and fully agreed upon by all researchers through a rigorous iterative process [43]. The themes were scrutinized by a team of subject experts, including clinical psychologists, physiotherapists, academics, and methodological experts, during the analytical process.

## 3. Results

### 3.1. Participants

All consecutive potential participants across the three programs in the study period, a total of 18 people, were contacted, and 11 participants consented to be interviewed. The reasons for non-participation were: no answer after three attempts to call (*n* = 4), too busy for the interview (*n* = 2), and self-reported inadequate English language skills (*n* = 1). Table 2 describes the demographic and clinical variables of these participants. The participants included six females and five males, who were aged between 26 and 72 years (median age = 55 years). Pain duration was variable, with 45% having pain for more than 5 years. The most frequently endorsed main pain site was back pain (*n* = 5) and the number of pain sites reported ranged from two to 11 sites. Eight participants (73%) reported they were unemployed due to pain, and 3 (27%) were retired from work by choice. Two participants had a worker’s compensation claim. On average, participants had moderate pain severity and interference, severe depression, severe anxiety, moderate stress, high pain catastrophizing, and severe pain self-efficacy. No participants reported feeling upset or distressed during the in-depth interviews.

Participants reviewed the transcripts of their interviews for accuracy. Two participants revised or added minor details to their interview transcripts.

### 3.2. Themes

Forty-seven codes were extracted from the transcripts supported by 163 illustrative quotes (Appendix A). No new codes were added after the tenth interview, and the researchers subsequently agreed that data saturation had been achieved after the eleventh interview. The codes were grouped into 20 subthemes from which four overarching themes emerged (Table 3). The titles of themes incorporate a rhetorical use of the phrase “but…” to emphasize the way that participants sometimes saw themselves as an exception to the rule. Therefore, we have expressed the themes using language similar to patient descriptions.

**Theme** **1.**
*The program overall was positive, but…*


Participants’ comments about the program overall were very positive, despite some participants demonstrating limited understanding or misinterpretation of key messages. Some of the specific comments suggested elements of lower value and impact, and that key messages of the program may have missed their mark. Many of the general comments suggested high levels of overall value and impact.


*“…It’s good. I’m pleased. I’m happy, more than happy.”*
*(Male, 72)*


*“…It did help me, and it helped me mentally as well.”*
*(Female, 69)*


*“…Overall, I’m ecstatic about how the program went.”*
*(Male, 26)*

The program was described as logical and credible, life-changing, something to be recommended, and exceeding initial expectations.


*“Everything made sense…”*
*(Male, 72)*


*“I can walk without pain. It’s a miracle that program.”*
*(Male, 72)*


*“I’ve spoken about the program to quite a lot of people & I would recommend it hands-down.”*
*(Female, 60)*


*“I was surprised. I didn’t expect it to be as good as it was.”*
*(Female, 51)*

For one participant, the program provided hope and optimism.


*“But with their help, I was able to see that there is a future providing I’m in the right mindset”*
*(Male, 26)*

Despite this overall positivity, some participants expressed negativity about their persistent pain.


*“…It just didn’t really work for me. I was like, I was always in pain.”*
*(Male, 63)*


*“My body is more flexible than before. But pain is not gone away...”*
*(Male, 47)*

It was also apparent that the beliefs of some participants in relation to the pathoanatomical basis of their pain were difficult to shift or that the overall program was not personally applicable to them. Other participants mentioned how information from their specialists conflicted with the overall program messages.


*“The specialist even said that nothing can be done…”*
*(Male, 47)*


*“I went to my neurosurgeon and he said that ‘you’ve got a bulging disc’. He said, ‘I’ll see you back for another op’ and as soon as you hear that you get scared…”*
*(Female, 51)*

**Theme** **2.**
*I valued my improved knowledge and understanding of pain, but…*


Participants valued their improved knowledge and understanding in relation to the key messages of the program, however, the extent to which the key educational messages were accepted varied. Participants described the messages as logical and credible, and value was attached to being more knowledgeable and better informed about pain in general. In some cases, the messages appeared to be wholly embraced.


*“It was good for me being a person who likes the information, to have that as background knowledge to understanding why I’m in pain and how to go about managing it. It really helped me.”*
*(Male, 26)*

For some participants, the messages were only partially or conditionally accepted, and in other cases, participants highlighted the value of key educational messages but rejected their relevance to their own specific situations.


*“My pain was completely different to other people’s pain and what I suffered was something different and that other people had different pain. But I could share and understand other people’s views. We shared a lot of things in common.”*
*(Female, 55)*


*“I can understand where they’re coming from, … but that hasn’t proven to be true in my case. I’ve had to have more medical intervention.”*
*(Female, 60)*

**Theme** **3.**
*I valued the stretching/relaxation/pacing/activity monitoring*


Several participants singled out individual aspects of the program they believed to be of particular benefit. These were typically the simple action-oriented components of the program, including the daily stretching exercise routine, relaxation training, and monitoring of self-pacing.


*“There were a couple of things I found really, really helpful & that was the easily adapted stretching & exercise regime each day.”*
*(Female, 60)*


*“I have a routine that helps me sleep at night, involving an hour of stretching that I adapted from the stretching part of the program. I do it every night before bed...”*
*(Male, 70)*


*“The meditation, separated from the context of some sort of religious spirituality, but just in terms of calming the nerves, is extremely important.”*
*(Male, 70)*


*“Start doing a little bit and increase more and more and more… You get a little bit of pain but don’t get put off, just keep going…”*
*(Male, 72)*


*“I could really see from the paperwork that we were filling out, how in black & white, I was progressing.”*
*(Female, 60)*

Participants highlighted how they tracked their progress on paper handouts, thereby demonstrating progress with these tasks. Some participants reported that they had adopted these routines into their daily lives and appeared to value the sense of achievement in doing so. Others who had not fully adhered with the recommendations about exercise, relaxation and pacing noted that they knew that they should have and believed that these factors were important in pain management.

**Theme** **4.**
*I valued being part of a supportive and understanding group*


Participants appeared to value the dynamics of a group-based program. Participants noted the support that they felt from the staff and peers within the group and noted the importance of being accepted and having their pain experience validated.


*“I thought J and T were both great. I thought J was inspirational. His energy and his passion and his belief in the program resonated not just with me, but everyone in our group.”*
*(Female, 51)*


*“It became possible to relate to the other individuals participating in the program. The small number of people, I got to know them almost every single one, personally.”*
*(Male, 70)*


*“[Outside of the group] Because we don’t look physically unwell, we haven’t got a broken leg with a cast on it, we’re not showing it & we’re not being believed.”*
*(Female, 60)*

The group setting also provided an opportunity for benchmarking progress with peers.


*“… Some of them obviously had more injuries than others… You can see by the end of that program how much they had progressed.”*
*(Male, 26)*

Participants also noted that despite the group setting, they felt that they were treated as individuals.


*“The program is good in that you can make it custom made to yourself.”*
*(Female, 60)*

## 4. Discussion

This study explored elements of the perceived value and impact of elements of a multidisciplinary group-based pain education program. Four themes emerged from these data. Firstly, while participants generally valued the program, there were elements of the program perceived to be negative, and there was a suggestion that some of the key messages may have missed their mark. Secondly, participants appeared to appreciate gaining new knowledge and education for its own sake, even when the knowledge was deemed not applicable to their situation and when the knowledge did not translate into changed beliefs, attitudes, or behaviors. Thirdly, participants singled out some of the simpler action-orientated components of the program as being highly valued, possibly because they involved clear and easy to implement recommendations. This contrasts with some of the more complex content involving changing beliefs, attitudes, and approaches. Finally, participants valued particular aspects of the group setting of the program particularly in relation to the overall mood and atmosphere. A unique finding from this study is the idea that clinicians can change a patient’s knowledge, beliefs, and attitudes regarding pain during pain programs, but it appears that patients must deem these changes to have personal relevance in order for the changes to have a significant impact on them.

The overall impressions and general appraisal by participants in this study were very positive, suggesting that the components of the Spark Pain Program were generally both impactful and valuable. This aligns with the large qualitative evidence base regarding the healthcare experience of people challenged by persistent pain, which has concluded that treating a patient with a sense that they are worthy of care and hearing their story is not an adjunct to, but integral to their care [47]. The mix of positive and negative feedback within individual interviews, and the fact that the interviewer identified herself as being independent from the program, suggests that the overall praise was a genuine reflection of value and impact in these participants. While the evidence of a direct association between broader constructs like patient satisfaction and clinical outcomes is uncertain [48], high levels of patient engagement, acceptance of key educational messages, shifting beliefs and attitudes, and behavioral change are all unlikely to occur in a group that is generally dissatisfied. 

Our results indicated that participants appreciated the group dynamics and aspects of the Spark Pain Program related to the mood and atmosphere rather than just content and delivery. Participants valued previously researched variables, such as the sense of validation and being believed [49], as well as the sense of emotional support they received from staff and peers [50]. Further to the current literature, participants also valued the sense of being treated as an individual, and the opportunity to benchmark their condition and their progress alongside “people like me”. This suggests that the soft skills of the program designers and facilitators contributing to the supportive and caring atmosphere of the program could be an important feature of program design. Future research could explore the value of ongoing follow-up support and fostering ongoing peer support to maintain this participant value in the longer term. 

The other features of the Spark Pain Program that were perceived as valuable and impactful were the simple action-orientated tasks such as stretching, exercises, relaxation, and self-monitoring activities. Participant reports also suggested that in many cases, these components resulted in lasting behavioral change. It is possible that the relative ease of implementation of these strategies influences their acceptability and perceived value. It is, therefore, possible that delivery of some of the more abstract key messages of the pain management program that involve shifting strong and entrenched beliefs might be more effective with better designed, action-orientated educational resources. 

Our results have also identified some potential weaknesses in traditional approaches to group-based pain management education and highlighted areas where key educational messages are misunderstood or not fully accepted. One recurrent theme was the acceptance of the inherent logic of some of the key messages while deeming the message to be inapplicable to the participant’s personal circumstances. If a patient observes themselves not improving and they also observe others in the group improving, attending the group program may reinforce the perception that “it’s good but not for me”. This is consistent with the Health Belief Model [51] for health-related behavior change, which emphasises the necessity of perceived applicability in terms of susceptibility, severity, benefits of change, and barriers as a precursor for behavioral change. The results of this study suggest that some of the messages with lower impact might benefit from delivery with an emphasis on the applicability or a “people like me” theme.

Our results suggest that the one particular key message that was not well accepted by all participants in this low-intensity program involved challenging the link between pathoanatomical findings and pain. Many participants, despite accepting other key messages such as “hurt versus harm” [52] and activity pacing, continued to use catastrophic pathoanatomical terms to describe the condition, and suggested that their diagnoses (e.g., disc protrusion) meant that a biopsychosocial [53] pain management approach [54] did not apply to their personal situation [55]. It appeared that a biomedical focus was tenacious and that any shift in beliefs was vulnerable to being contradicted and undermined by other health professionals. This finding is supported by previous qualitative research following pain science education, where varying degrees of reconceptualization were observed, from zero to almost complete [56]. While face-to-face interviews may have increased the depth of the responses, and these results highlight the need for future quality improvement innovations in pain management to develop more effective and convincing ways of delivering this important message. It also suggests that improved pain science education [32,33] of the general public and health professionals regarding the limitations of a biomedical model for chronic pain is required.

This qualitative study has several strengths and limitations. A strength of this study was the large variations in pain experience among the participants in terms of pain duration, pain sites, work status, and compensation status, capturing individual variability in experiences of pain [57]. A further strength is that the interviewer was independent of the program. One limitation of this study is the potential lack of transferability. This study only included adults from New South Wales, Australia, who spoke English. It is unknown if the results would be transferable to other cultures and age groups. A further limitation is that the structured telephone interviews were relatively brief and “to-the-point”, and although they are convenient, telephone interviews have potential biases such as difficulty with developing rapport and the potential for missing visual cues [58]. Therefore, the themes resulting from this research would be suitable to investigate in greater depth in future longer duration face-to-face interviews using triangulation [59]. Further detailed analysis of the meaning of the responses (asking participants why they felt the way that they did about the impactful and valuable factors) is another direction for further research, as is co-design, which would use our findings for quality improvement using different methodologies.

## 5. Conclusions

These findings provide insights into aspects of the Spark Pain Program that are linked to patient values and impacts, in addition to a better understanding of aspects of the program that might be missing their mark. It appears that patients in pain programs must determine that changes in knowledge, beliefs, and attitudes, are personally relevant in order for the changes to have a significant impact on them. Perceived value and impact may relate closely to the perceived ease of implementation of different strategies and the action-oriented nature of educational resources. This new knowledge provides opportunities for quality improvement as well as opportunities for implementing innovations in educational design into group-based pain management programs.

## Figures and Tables

**Table 1 medicina-57-00046-t001:** Guiding script for the interviews.

**Demographic and clinical information:**
Age Gender Pain characteristics: ○ Sites ○ Duration ○ Intensity ○ Interference When did you finish the Spark Pain Program?
**Open ended questions:**
What was your overall impression of the Spark program? (Best bits, worse bits, what did they do really well, what did you find not as useful) ○ Prompts: was it helpful, or unhelpful? Did you learn anything that you didn’t know before? ○ Has this been helpful? Why is this important? Is there anything that you are thinking about differently? Is there anything that you now do differently? Any new habits or routines? Anything you have stopped doing or changed dramatically? What advice would you give someone who has the same condition that you have?

**Table 2 medicina-57-00046-t002:** Demographics and clinical variables of the participants at baseline.

Categorical Variables	*n* (%)
Gender	
Male	5 (45)
Female	6 (55)
Pain duration	
3 to 12 months	1 (9)
1 to 2 years	3 (27)
2 to 5 years	2 (18)
More than 5 years	5 (45)
Main pain site	
Back	5 (45)
Hip	2 (18)
Neck	1 (9)
Head	1 (9)
Shoulder	1 (9)
Knee	1 (9)
Work status	
Unemployed due to pain	8 (73)
Not working by choice (retired)	3 (27)
Compensation claim status	
Yes	2 (18)
No	9 (82)
**Continuous Variables**	**Mean (SD)**
Age in years, range, median	55 (14), 26 to 72, 55
Number of pain sites, range	5.7 (2.5), 2 to 11
BPI Severity /10	6.5 (2.0) “Moderate”
BPI Interference /10	6.8 (2.0) “Moderate”
DASS Depression /42	23.4 (9.5) “Severe”
DASS Anxiety /42	16.0 (11.5) “Severe”
DASS Stress /42	24.0 (9.8) “Moderate”
PCS /52	29.7 (14.2) “High”
PSEQ /60	19.3 (14.6) “Severe”

BPI, Brief Pain Inventory [44]; DASS, Depression, Anxiety and Stress Scale Lovibond and Lovibond 1995); PSEQ, Pain Self-Efficacy Questionnaire [45]; PCS, Pain Catastrophizing Scale [46]. For the PSEQ assessment tool, a higher score indicates an increased ability to perform activities despite the pain. For the BPI, DASS, and PCS, a higher score is a worse clinical outcome.

**Table 3 medicina-57-00046-t003:** Subthemes and themes emerging from the data.

Subthemes	Themes
Overall very positive	The program overall was positive, but…
Exceeded expectations	
Recommend to others	
Motivating, provided a sense of hope	
Logical and made sense	
Transformative, life changing	
I still have pain	
It didn’t apply to me	
Inconsistent with specialists	
Total acceptance of key messages	I valued my improved knowledge and understanding of pain, but…
Partial or conditional acceptance of key messages	
Rejection of key messages	
Stretches	I valued the stretching/relaxation/pacing/activity monitoring
Relaxation	
Pacing and activity monitoring	
Staff support	I valued being part of a supportive and understanding group
Peer support	
Validation, being believed	
Benchmarking	
Being treated as an individual	

## Data Availability

De-identified transcripts, and the unpublished protocol, will be made available upon reasonable request from Joshua Pate (https://orcid.org/0000-0002-1049-3916) until 2025 as per ethical approval. Reuse without ethical approval is not permitted. A data-sharing agreement will require a commitment to using the data only for specified research purposes, to securing the data appropriately, and to destroying the data after a nominated period.

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
