# Peer review of "The Importance of Perceived Relevance: A Qualitative Evaluation of Patient’s Perceptions of Value and Impact Following a Low-Intensity Group-Based Pain Management Program"

_medicina, 2021, doi:10.3390/medicina57010046_

Round 1
Reviewer 1 Report
The article was relevant and well-written. I have some concerns, but I think the authors will be able to address all of them.
“Approximately 9.9 million workdays absent were attributed to persistent pain in Australia in 2006 [4]”. This is based on data from 14 years ago, can you update this citation?
Each week involves pain science education (20min), relaxation (10min) exercise (30min), problem-solving (30min), stretching (20min) and goal setting (10min). Please, include more information about the program since it is the key to the interviews.
Regarding the participants, how many people participated in the program? 18? Do you think those 11 patients are a representative sample of the patients that were involved in the program? Do you think there could be some bias? For instance, those patients who were more satisfied with the program might be more likely to consent to be interviewed than those who did not enjoy it.
I think the discussion section should be improved. What is the main contribution of this article? The organization of the discussion section is also unclear. The last paragraph may be better placed before talking about strengths and limitations, after the interesting paragraph about potential weaknesses in traditional approaches.
I miss some practical recommendations in the discussion section. It is true that they can be extracted from the information provided in different paragraphs, but I recommend pointing them clearly.
It is not easy to value the discussion section since the available information about the program is vague. Please, include more information about the pain management program.
Author Response
Thank you for your review. Please see pages 1-3 of the attached document for point-by-point responses to your comments.

Reviewer 2 Report
This qualitative study took place in a “low intensity” pain management program in Australia; the study objective was to identify which dimensions of the program were perceived as impactful by patients. Though well-written, this article does not provide sufficient contribution to the existing qualitative literature in the field. The qualitative design contains important weaknesses due to the short duration of interviews, which does not enable in-depth qualitative analysis and production of robust findings. The presentation of results also contains logical deficiencies.
Detailed comments:
Introduction:
In the introduction, more information is needed regarding the specific type of intervention studied (“lower intensity programs”) as well as the Australian context of pain care organization within which this intervention is deployed. This specific context is important to assess the study’s transferability to other types of programs and local contexts.
The rapid overview of previous qualitative studies in the field is incomplete. For decades, an abundant literature of qualitative studies has documented various aspects of the chronic pain experience and management from the perspectives of patients and providers.
Methods:
Though the qualitative methods are reported adequately using the COREQ guidelines, the design of the study presents major problems. The main issue is the extremely short duration of interviews (11 minutes on average). This is not acceptable in a qualitative study. More in-depth narrative are required to collect data that can be used in a proper qualitative analysis.
This weakness is reflected in the results, as the authors can only use very short quotes of little interest in terms of qualitative content. In-depth analysis of the meaning of the intervention to participants is impossible in these conditions.
In addition, short interviews assessing patient satisfaction are likely to lead to several bias during data collection, as a relationship of trust between the interviewer and the participant cannot be built. This reduces the validity and relevance of findings regarding participants’ satisfaction.
In addition, it is improper to refer to a “semi-structured” interview design in the case of such short interviews. Semi-structured interviews usually last one hour or more. It is unlikely that a semi-structured interview could last 10 minutes and meet its goals of collecting in-depth responses from participants. Using the interview guide like a questionnaire is not recommended. Referring to a “structured interview” design would be more appropriate to describe what the authors effectively did (asking several questions to obtain short responses). Furthermore, a semi-structured interview guide needs to contain open-ended questions only, which is not the case in the guide provided by the authors (5 close-ended questions).
The reference to grounded theory is inappropriate as there is no theory-building in this study. Authors should refer to thematic analysis only.
The described means for assessing data saturation seem insufficient. Saturation cannot be ascertained with only one interview presenting redundancy with the previous ones. Authors may want to refer to data sufficiency or information power instead of saturation. Please see the references below:
Malterud K, Siersma VD, Guassora AD. Sample Size in Qualitative Interview Studies: Guided by Information Power. Qual Health Res. 2016;26(13):1753-1760. doi:10.1177/1049732315617444
Varpio L, Ajjawi R, Monrouxe LV, O’Brien BC, Rees CE. Shedding the cobra effect: problematising thematic emergence, triangulation, saturation and member checking. Medical Education. 2017;51(1):40-50. doi:10.1111/medu.13124
Eligibility criteria and dates when the interviews were conducted should be specified.
Results/discussion:
The insufficient duration of interviews has a negative impact on the results. The authors are unable to provide detailed analysis. The findings are supported by very short quotes, which present little relevance and evidence of the authors’ interpretations. The contribution of these results to qualitative research in the field is therefore very small.
The organizing of themes contains some logical flaws. It is unclear why contradictory results (i.e. positive and negative experiences) are combined in one theme (it is the case in Theme 1 and Theme 2).
Furthermore, some aspects of Theme 1 and Theme 2 are redundant:
e.g: Theme 1 (p 6 line 172): “It was also apparent that the beliefs of some participants in relation to the pathoanatomical basis of their pain were difficult to shift or that the messages were not personally applicable to them.”
Theme 2 (p 6 line 187) : “participants highlighted the value of key messages but rejected their applicability to their own situations.”
The titles of Themes 1 and Theme 2 ending with “but…” are not explicit and contribute to hiding information about participants’ negative perceptions.
Theme 3 provides a list of aspects that patients reported being valuable. It is of poor interest in a qualitative perspective, as these dimensions and their meaning to participants are not analyzed in detail (dimensions are only named). It is likely that the poor quality of data due to short interviews will not enable further analysis.
Theme 4 seems more interesting as it may reflect an emerging finding (the importance of group support). However, a more detailed analysis would have been welcome in this section too, which is probably impossible as well.
The discussion provides some interesting issues for intervention (e.g. importance to emphasize applicability of key messages in order to engage participants in the program), however the study’s novelty and original contribution to the literature in the field of multidisciplinary pain treatment remains unclear.
Author Response
Thank you for your review. Please see pages 4-10 of the attached document for point-by-point responses to your comments.

Round 2
Reviewer 1 Report
The authors have addressed all my concerns